Applied and Environmental Science
# Assessing Biodegradability of Chemical Compounds from Microbial Community Growth Using Flow Cytometry

B. D. Özel Duygan,[a]* S. Rey,[b] S. Leocata,[c] L. Baroux,[c] M. Seyfried,[b] J. R. van der Meer[a]

aDepartment of Fundamental Microbiology, University of Lausanne, Lausanne, Switzerland
bBiotechnology and Natural Process Development Department, Firmenich SA, Geneva, Switzerland
cInnovation in Analytical Chemistry Department, Firmenich SA, Geneva, Switzerland

**ABSTRACT** Compound biodegradability tests with natural microbial communities form an important keystone in the ecological assessment of chemicals. However, biodegradability tests are frequently limited by a singular focus either on the chemical and potential transformation products or on the individual microbial species degrading the compound. Here, we investigated a methodology to simultaneously analyze community compositional changes and biomass growth on dosed test compound from flow cytometry (FCM) data coupled to machine-learned cell type recognition. We quantified the growth of freshwater microbial communities on a range of carbon dosages of three readily biodegradable reference compounds, phenol, 1-octanol, and benzoate, in comparison to three fragrances, methyl jasmonate, myrcene, and musk xylene (as a non-biodegradable control). Compound mass balances with between 0.1 to 10 mg C · liter$^{-1}$ phenol or 1-octanol, inferred from cell numbers, parent compound analysis, and $CO_2$ evolution, as well as use of $^{14}$C-labeled compounds, showed between 6 and 25% mg C · mg C$^{-1}$ substrate incorporation into biomass within 2 to 4 days and 25 to 45% released as $CO_2$. In contrast, similar dosage of methyl jasmonate and myrcene supported slower (4 to 10 days) and less (2.6 to 6.6% mg C · mg C$^{-1}$ with 4.9 to 22% $CO_2$) community growth. Community compositions inferred from machine-learned cell type recognition and 16S rRNA amplicon sequencing showed substrate- and concentration-dependent changes, with visible enrichment of microbial subgroups already at 0.1 mg C · liter$^{-1}$ phenol and 1-octanol. In general, community compositions were similar at the start and after the stationary phase of the microbial growth, except at the highest used substrate concentrations of 100 to 1,000 mg C · liter$^{-1}$. Flow cytometry cell counting coupled to deconvolution of communities into subgroups is thus suitable to infer biodegradability of organic chemicals, permitting biomass balances and near-real-time assessment of relevant subgroup changes.

**IMPORTANCE** The manifold effects of potentially toxic compounds on microbial communities are often difficult to discern. Some compounds may be transformed or completely degraded by few or multiple strains in the community, whereas others may present inhibitory effects. In this study, we benchmark a new method based on machine-learned microbial cell recognition to rapidly follow dynamic changes in aquatic communities. We further determine productive biodegradation upon dosing of a number of well-known readily biodegradable tester compounds at a variety of concentrations. Microbial community growth was quantified using flow cytometry, and the multiple cell parameters measured were used in parallel to deconvolute the community on the basis of similarity to previously standardized cell types. Biodegradation was further confirmed by chemical analysis, showing how distinct changes in specific populations correlate to degradation. The method holds great promise for near-real-time community composition changes and deduction of compound biodegradation in natural microbial communities.

**Citation** Özel Duygan BD, Rey S, Leocata S, Baroux L, Seyfried M, van der Meer JR. 2021. Assessing biodegradability of chemical compounds from microbial community growth using flow cytometry. mSystems 6: e01143-20. https://doi.org/10.1128/mSystems .01143-20.

Address correspondence to B. D. Özel Duygan, birgeozel@gmail.com, or J. R. van der Meer, janroelof.vandermeer@unil.ch.

*Present address: B. D. Özel Duygan, Institute for Microbiology CHUV, Lausanne, Switzerland.

Rapid flow cytometry analysis coupled to machine learned cell type recognition unveils specific population changes of freshwater communities exposed to a variety of model chemical assaults, and is linked to compound biodegradation.

**KEYWORDS** biodegradation, flow cytometry, freshwater, machine learning, microbial communities

Industrial chemicals are essential ingredients of modern societies, accounting for more than 95% of all produced goods (https://www.oxfordeconomics.com/recent-releases/the-global-chemical-industry-catalyzing-growth-and-addressing-our-world-sustainability-challenges). Despite providing many benefits, their release over the course of their life cycle may cause negative effects on human health and the environment (1, 2). A crucial aspect of environmental risk analysis therefore includes an assessment of the biotransformation potential of a compound after its introduction into an environmental compartment (3). Natural microbial communities play an important role in both spontaneous compound transformation and mineralization, thereby using the compound as a whole or in part as a carbon or nutrient substrate for growth and/or for energy generation (4, 5).

Despite a wealth of information on compound transformation pathways and transformation kinetics, the physiology, biochemistry, and genetics of individual microorganisms carrying out transformation reactions, it is still very challenging to obtain an integrative assessment of a compound's transformation fate and its corresponding effects within a typical complex microbial ecosystem and its environmental conditions. Biodegradation tests follow a clear tiered standardized framework, which involves pure compound and standardized microbial biomass sources and standardized pass/fail options to assess the potential and ease of biodegradability (3, 6). These tests are compound oriented and focus on parent compound disappearance (by total dissolved organic compound measurements), mineralization outputs such as $CO_2$ evolution, or trace compound kinetics (3). Despite its analytical value, the standardized ready biodegradability test approach has also been criticized as being too rigid, leading to potentially high rates of false negatives (7), and use of high compound test concentrations (10 to 400 mg C $\cdot$ liter$^{-1}$) (3), with difficulties for hydrophobic and volatile test compounds (8), and largely for treating the inoculum source as a black box (3). Ready biodegradability tests assume that sufficient or representative biodegradation capacity is present in the natural source microbial community (5), whereas this is arguably quite variable (7, 9, 10) and may depend on source, seasonality, and inoculum concentration (3). To provide a more realistic perspective on compound biodegradation in an environmental compartment, it would thus be important to identify the activity and growth of primary degraders, and potentially further trophic relationships. This has been achieved by stable isotope probing of carbon flow and identification of [13]C-enriched populations from 16S rRNA amplicon analysis (11) or metagenomic sequencing (12). Although extremely powerful, both DNA- and RNA-stable isotope probing are limited by the availability of [13]C-labeled substrate and are cumbersome in terms of dense and near-real-time sample analysis. As a potential alternative, we focused here on flow cytometry (FCM). The key advantage of FCM is that it can rapidly and accurately quantify microbial cell numbers in suspension across several orders of magnitude (13). A current major difficulty in FCM analysis is the simultaneous detection of community compositional changes. However, recent advances with machine-learned algorithms have shown meaningful deconvolution of complex communities into cell types (14, 15) and distinction of temporal trends or substrate effects (16–18).

The basic assumption for FCM testing of chemical biotransformation is that if *de novo* introduced compounds in a microbial ecosystem are metabolizable by one or more members of that ecosystem, it may lead to cell division and growth of the community and become detectable by FCM. This is because during organic compound transformation, part of the compound's carbon (and possibly other atoms, such as N) and/or released metabolic energy may become available for the production of new biomass (i.e., new cells). An increase in microbial cell numbers in a suspension could thus potentially be used as a readout for the compound's transformation, instead of or in addition to measuring compound disappearance or chemical transformation

products. Compound dosage to a microbial community would thus be expected to result in measurable increase in cell numbers if such a compound is productively transformed into biomass during the assay period. FCM is accurate even at relatively low carbon dosages (e.g., $10\,\mu g$ acetate-C $\cdot$ liter$^{-1}$ led to a measured increase of $10^5$ cells $\cdot$ ml$^{-1}$ [19]). One might thus envision measuring the number of newly formed cells at the expense of dosed test compounds. In practice, several potential confounding factors may impede correct inference of compound biotransformation from increasing cell numbers in an assay, such as growth on available background carbon, carbon recycling within microbial food webs, or autotrophic growth. For example, even the purest aqueous medium contains 10 to $100\,\mu g \cdot$ liter$^{-1}$ of assimilable organic carbon (AOC), sufficient to generate $\sim 10^8$ to $10^9$ bacterial cells $\cdot$ liter$^{-1}$ (20–22), thus potentially limiting the net growth distinction to an additional test compound dosed to the system (23). If, however, one could simultaneously unravel subgroups within the microbial community, one could potentially uncover those that react specifically to the dosed compound.

The goal of the current study was thus to evaluate a methodological framework based on FCM cell counting coupled to machine learning algorithms to classify cells from multidimensional FCM data and uncover community composition changes as a result of substrate dosing. FCM community counting had previously shown promising results to detect cell density increases at the expense of dosed individual compounds (23, 24). More recently, an artificial neural network-based classifier algorithm was developed to attribute cells from microbial community samples to one of 32 standard categories on the basis of the closest FCM multidimensional resemblance (15). With this basis in hand, we aimed here to validate microbial community growth at different starting cell densities and across a wide range of concentrations (from 0.1 mg C $\cdot$ liter$^{-1}$ to 1 g C $\cdot$ liter$^{-1}$), for three well-known readily biodegradable compounds (benzoate, phenol, and 1-octanol), two fragrances known to be readily biodegradable (methyl jasmonate and myrcene), and the nonbiodegradable compound musk xylene as a negative control, as listed by Comber and Holt (25). As a relevant inoculum, we used a freshwater microbial community, sampled at different seasons from the same location. We benchmarked the test by quantifying mass balances from community growth, $CO_2$ evolution, and parent compound disappearance, as well as from $^{14}$C-labeled substrates (phenol and 1-octanol). Community composition changes upon compound incubation were assessed from FCM-deconvoluted class attributions and, in selected examples, compared to 16S rRNA gene amplicon analysis of community composition, in order to detect general trends and identify enriched populations driving the primary degradation.

## RESULTS

**Microbial community growth on individually dosed compounds at different concentrations.** In order to assess biotransformation potential of compound and biodegradation capacity of natural microbial communities, we dosed individual substrates at various carbon concentrations (0.1 mg C $\cdot$ liter$^{-1}$ to 1 g C $\cdot$ liter$^{-1}$) and quantified community cell growth by FCM within a time period of up to 10 days (depending on the experiment; see Table S1 in the supplemental material). As a source of freshwater microbial community, we sampled and recovered cells from Lake Geneva water, which were resuspended to between $10^4$ to $10^6$ cells $\cdot$ ml$^{-1}$ in artificial lake water (ALW) medium (see Table S2 in the supplemental material). Growth on ALW without any added carbon substrate functioned as a negative control for background growth.

Addition of any of the three readily biodegradable standard reference compounds, benzoate, phenol, and 1-octanol caused rapid (2 to 4 days) community size increase, indicating growth of one or more species in the freshwater community at the expense of the added carbon (Fig. 1A). Community size reached a maximum within 3 to 5 days, depending on substrate concentration (Fig. 1A), but after that it frequently decreased (see Fig. S1 in the supplemental material), which might be due to cell clumping, lysis, or predation within the community. Dosages of 1 g C $\cdot$ liter$^{-1}$ resulted in less community growth than that extrapolated from lower substrate concentrations, indicating toxicity and growth inhibition (Fig. 1A). The lowest discernible community growth on

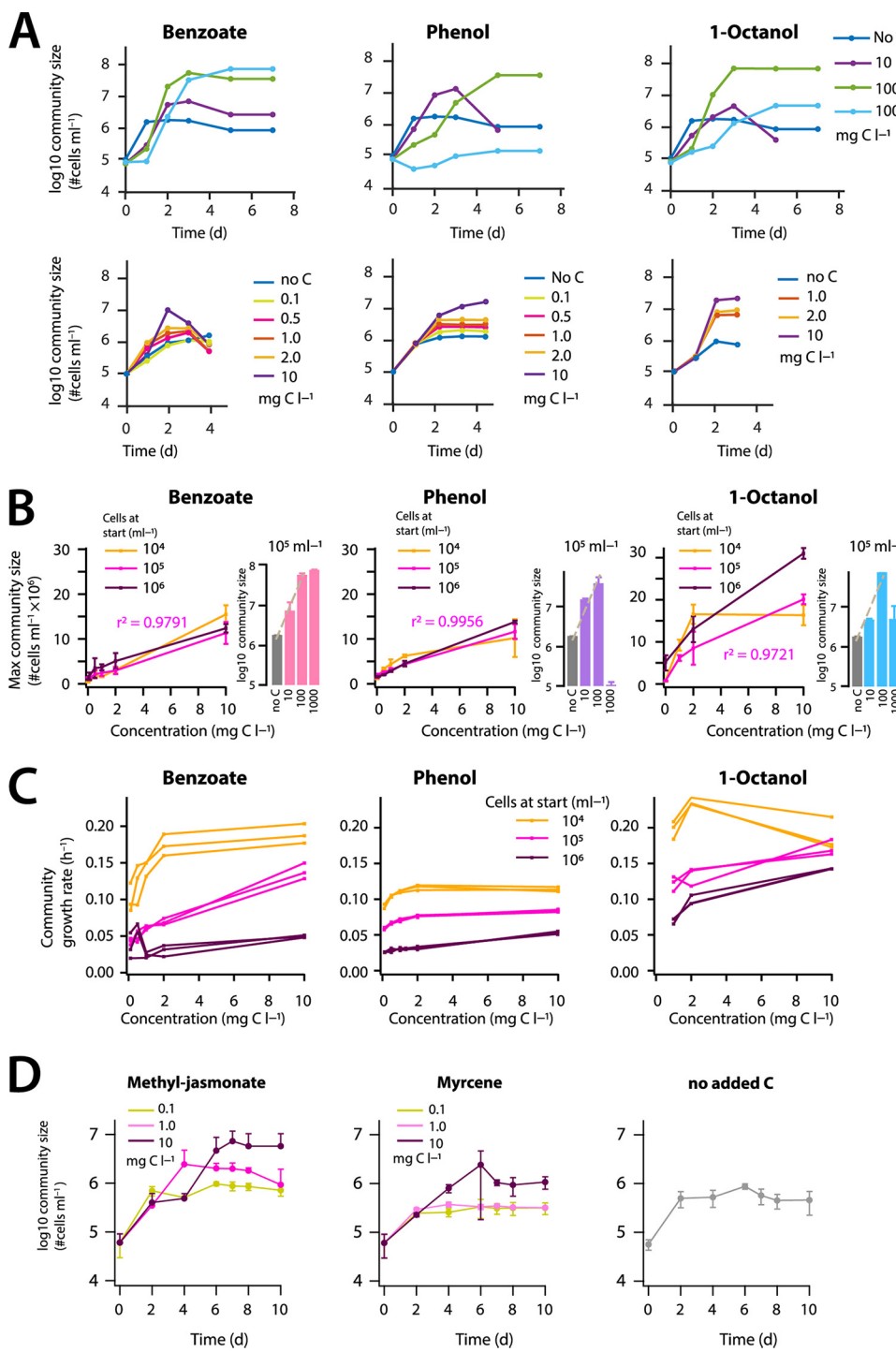

**FIG 1** Growth properties of freshwater microbial community with selected dosed compounds at different concentrations. (A) Community growth at $10^5$ cells · ml$^{-1}$ at start with benzoate, phenol, or 1-octanol at various initial concentrations (upper series: high concentration range; lower series: low concentration range). Symbols show mean community sizes from triplicate assays in cells · ml$^{-1}$ measured by flow cytometry (FCM), plotted on a log$_{10}$ scale (error bars omitted for clarity). (B) Effect of initial starting community cell number on the maximum community size for three compounds at different concentrations. Data points show the mean community sizes ± one standard deviation (SD) from triplicate assays measured by FCM, on a linear scale (lower concentration range) or log scale (bars for higher concentration range). Linear regression coefficients ($r^2$) calculated for the $10^5$ cells · ml$^{-1}$ starting community density. (C) Relationship of the community-specific growth rate as a function of compound, concentration, and starting cell density. Data points show individual replicates, grouped arbitrarily by connecting lines. (D) Community growth on methyl jasmonate or myrcene at different dosages. Data points show the mean community sizes ± one SD (error bars) from triplicate assays measured by FCM.

added substrate in comparison to growth in no-carbon controls was at 0.1 mg C · liter$^{-1}$, but this depended on the AOC background of the ALW. From the observed growth in no-carbon controls across multiple experiments, we estimated background AOC levels of between 71 to 134 $\mu$g · liter$^{-1}$ (assuming that 1 $\mu$g of AOC is sufficient to allow growth of 10$^7$ bacterial cells · liter$^{-1}$) (13, 21, 22).

Similarly to what is typically observed for pure culture growth, the extent of the freshwater microbial community growth was dependent on the added substrate concentration. An almost linear increase in maximum community size as a function of added substrate concentration was observed for benzoate and phenol, up to 100 mg C · liter$^{-1}$, which was independent of the community starting cell density (10$^4$, 10$^5$, and 10$^6$ cells · ml$^{-1}$, Fig. 1B). In contrast, community growth was nonlinear for 1-octanol and was dependent on the community starting cell density (Fig. 1B, 1-octanol). This suggested that, depending on the concentration of 1-octanol, different microbial populations (or subgroups) in the community may be responsible for its degradation (see below).

Apparent maximum community growth rates were generally highest for the lowest starting cell density (10$^4$ cells · ml$^{-1}$), and increased in the range of 0.1 to 10 mg C · liter$^{-1}$ up to 0.20 h$^{-1}$, depending on the substrate (Fig. 1C). It should be noted that community growth is the sum of that of all microbial populations within the community, which may grow at different rates and to different extents and be subject to further population control mechanisms (see further below).

In contrast to the three readily biodegradable standard reference compounds, community growth on the two fragrances, methyl jasmonate and myrcene, was slower (4 to 10 days) and less pronounced (Fig. 1D). No net community growth in comparison to the no-carbon control (ALW) was observed at 0.1 and 1.0 mg C · liter$^{-1}$ for the first 100 h of incubation with methyl jasmonate and throughout the complete duration with myrcene (Fig. 1D). Community growth on methyl jasmonate increased after 100 h for dosages of 1.0 and 10 mg C · liter$^{-1}$ (Fig. 1D). Collectively, these results indicated that community size increases in comparison to a no carbon control can be deployed to infer compound biodegradability in the range of 1 to 10 mg C · liter$^{-1}$ by natural freshwater communities at low starting suspension densities (e.g., ~10$^5$ cells · ml$^{-1}$).

**Effect of background carbon on community growth.** Despite the background growth on AOC in no-carbon controls, it is conceivable that compound addition would enhance the community's capacity to multiply, for example, by causing cell death and reutilization of released carbon from dead cells. This increased background carbon or cell growth on carbon released from dead cells might confound the interpreted net growth at the expense of dosed test compounds.

To address how much background growth would occur from dead cell biomass, we repeated incubations of 10$^5$ cells · ml$^{-1}$ at the start with 10 mg C · liter$^{-1}$ of phenol in the presence or absence of 25% dead cells (killed by pasteurizing), in comparison to the no-carbon control. In case of the no-carbon control, including dead cells yielded on average 25% (range, 21 to 29%) more cells in the community at the same sampling time point (Fig. 2), although this was a statistically nonsignificant increase in pairwise comparisons ($P > 0.05$; two-sided $t$ test). In the presence of 10 mg C · liter$^{-1}$ of phenol and 25% dead cells, there were on average 10% less cells (range, 0 to 80%) at the same sampling time point, but this was also a statistically nonsignificant decrease ($P > 0.05$; two-sided $t$ test) (Fig. 2). The fraction of dead cells (propidium iodide positive) in the freshwater community suspension at start was 1 to 2%. This experiment thus indicated that it is unlikely that net growth observed at low added substrate concentrations (0.1 to 10 mg C · liter$^{-1}$) is due to growth on dead biomass generated by the added substrate.

**Mass balance analysis of compound biotransformation.** In order to understand how much of the added carbon substrate was transformed into community biomass, we measured substrate utilization, $CO_2$ evolution, and biomass formation in an independent series of experiments with 10 mg C · liter$^{-1}$ dosages of phenol, 1-octanol,

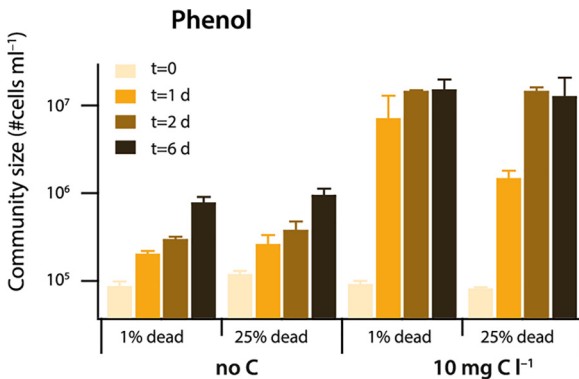

**FIG 2** Effect of potential background growth on carbon released from dead cells. Bars show the mean community size (+1 SD) among four biological replicate assays sampled at the indicated time intervals, in presence or absence of added phenol (to 10 mg C · liter$^{-1}$), and containing or not 25% dead (pasteurized) cells. Initial cell density, $10^5$ cells · ml$^{-1}$. Note that the community cell density is plotted on a $\log_{10}$ scale. Mean values at the same time points in the absence or presence of 25% dead cells are not statistically significantly different ($P$ values $> 0.05$; two-tailed $t$ test).

methyl jasmonate, and myrcene, in comparison to the nondegradable compound musk xylene and a no-carbon control.

As before, communities amended with phenol or 1-octanol rapidly increased in size (Fig. 3A and B, gray bars). This was accompanied by substrate disappearance down to concentrations below the detection limit of the chemical analytics after 48 h, and it correlated with an increase of $CO_2$ evolved over time (Fig. 3A and B). The percentage

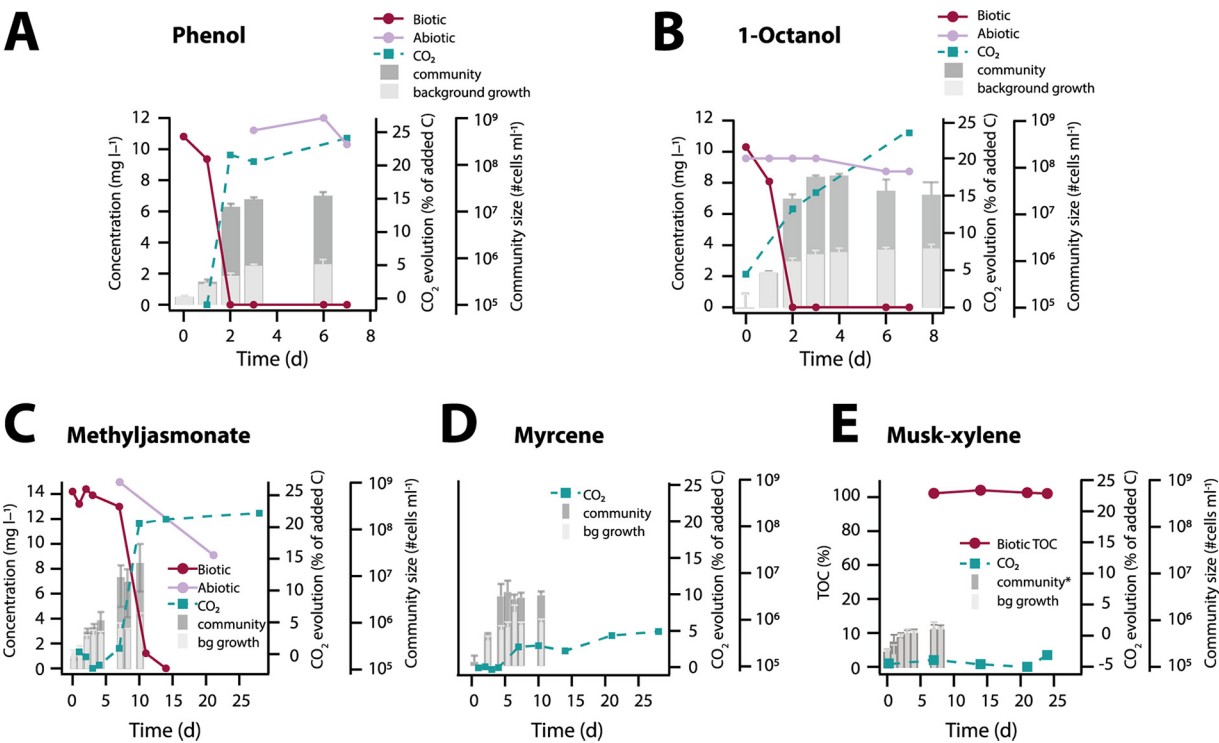

**FIG 3** Mass balance analysis for biotransformation of dosed compounds at 10 mg C · liter$^{-1}$ to freshwater community. (A) Disappearance of phenol (red points and connecting line) and concomitant $CO_2$ production (green squares; note separate scale) and community growth (gray bars; separate scale, means of biological triplicates plus 1 SD). Comparison: phenol concentration in an abiotic control (purple points). (B to E) As in panel A, but for 1-octanol, methyl jasmonate, myrcene, and musk xylene, respectively. Note the longer incubation times for experiments shown in panels C to E. Further, note that musk xylene was measured as total organic carbon (TOC), whereas extraction efficiency for myrcene was insufficient to properly calculate actual remaining compound. "bg growth," cell growth (per ml) on medium alone (i.e., no-carbon control). *, not significantly different from bg growth.

**TABLE 1** Carbon mass recovery in freshwater microbial community incubations

| Compound | Dosage (mg C · liter$^{-1}$) | C in biomass (%)[a] | C in $CO_2$ (%)[b] | Parent compound (%)[c] |
|---|---|---|---|---|
| Phenol | 10 | **15.9–24.5** | 23.9 | 0 |
| 1-Octanol | 10 | **7.3–27.4** | 31.9 | 0 |
| Methyl jasmonate | 10 | **2.6–5.1** | 22.2 | 0 |
| Myrcene | 10 | **4.0–6.6** | 4.91 | ND |
| Musk xylene | 10 | 0.66 ± 0.23 | −4.94 | 100[d] |
| No-carbon control | | 0.20 ± 0.05 | ND | |

[a]Range of mean summed biomass at maximum observed community sizes calculated using CellCognize. Values in bold typeface are significantly different from those of the no-carbon control (paired two-tailed $t$ test, $n = 3$, $P < 0.05$).
[b]Maximum measured $CO_2$ evolution by headspace within 28 days.
[c]Minimum residual percentage of parent compound measured by gas chromatography (GC). ND, not determined.
[d]As a percentage of total organic carbon analysis compared to that at $t = 0$.

of carbon mass released as $CO_2$ varied between 22 and 32% for both compounds during the 28-day incubation period (Fig. 3A and B and Table 1). Community growth arrest and compound disappearance after 2 to 4 days for incubations with phenol or 1-octanol suggested that all dosed substrate had been metabolized (Fig. 3A and B, gray bars). Methyl jasmonate took longer (14 days) to completely disappear in lake water community incubations, but compound decrease correlated again to an increase of community size (Fig. 3C). The percentage of carbon mass released as $CO_2$ was in the range of what was observed for phenol and 1-octanol (Fig. 3C and Table 1). In contrast, dosage of 10 mg C · liter$^{-1}$ of myrcene resulted in poorer $CO_2$ evolution (max 5%; see Table 1) and a smaller community size increase (Fig. 3D). Finally, no net growth of the freshwater community was observed with a dosage of 10 mg C · liter$^{-1}$ musk xylene (Fig. 3E and Table 1).

To translate the community cell number increase into cell biomass (i.e., as mg C), we deployed a recently developed flow cytometry machine learning pipeline for microbial cell type classification (CellCognize) (15). This pipeline assigns cells in unknown samples to a set of 32 predefined standard characterized microbial cell types or microbeads on the basis of their closest multidimensional resemblance of measured flow cytometry parameters. By using an estimated dry carbon mass of the standards based on their volume (15), one can then sum up all the assigned cell masses in the sample. For phenol and 1-octanol, this resulted in carbon mass incorporation into biomass of between 7.3 and 27.4%, depending on the sampling time of community size measurement (Table 1, days 2 to 4). Considering the proportion of developed $CO_2$, between 51.6 and 60.2% (for phenol) and 40.6 and 60.8% (for 1-octanol) of added C was thus not recoverable and may have been transformed into extracellular material that was not detectable as such in our assay (see below). In case of methyl jasmonate and myrcene, maximally between 2.6 and 6.6% of added substrate C was recovered in the community biomass (Table 1). The maximum observed $CO_2$ releases of 22.2% and 4.9% $CO_2$, respectively, for methyl jasmonate and myrcene (Table 1), were thus suggestive for their partial biodegradation in the freshwater microbial community assays. Incubations with musk xylene did not yield any significant net community biomass formation (Table 1), which, together with the lack of measurable $CO_2$ evolution and remaining presence of parent compound throughout the experiment are in agreement with its reported nonbiodegradability (8).

To corroborate community biomass yields further, we repeated freshwater community incubations with $^{14}$C-labeled phenol (uniformly labeled) and 1-octanol (end labeled) at substrate concentrations of 0.1, 1, and 10 mg C · liter$^{-1}$. As before, these substrate concentrations led to rapid community growth and stationary phase within 3 to 4 days, which was indicative for complete substrate utilization as in previous experiments (see Fig. S2 in the supplemental material). The distribution of $^{14}$C label was then

mSystems®

**TABLE 2** Recovery of $^{14}C$ label in community substrate incubations after 3 days

| Compound[a] | Dosage[b] (mg C · liter$^{-1}$) | % Radioactivity recovered in[c] | | |
|---|---|---|---|---|
| | | Filtered biomass[d] | Trapped gas phase[e] | Filtrate[f] |
| Phenol (set 1) | 0.1 | 8.4 ± 4.9 | 35.6 ± 5.3 | 53.3 ± 2.6 |
| | 1.0 | 8.5 ± 0.6 | 36.2 ± 2.6 | 51.7 ± 8.0 |
| | 10 | 15.8 ± 4.9 | 44.5 ± 6.5 | 33.1 ± 6.9 |
| Abiotic control | | 1.3 ± 0.1 | 3.5 ± 0.0 | 90.2 ± 0.2 |
| Phenol (set 2) | 0.1 | 12.4 ± 1.0 | 29.6 ± 1.0 | 53.9 ± 9.0 |
| | 1.0 | 11.5 ± 1.5 | 29.7 ± 1.5 | 54.0 ± 6.9 |
| | 10 | 11.9 ± 4.0 | 43.4 ± 6.6 | 40.3 ± 1.3 |
| Abiotic control | | 1.9 ± 0.1 | 1.8 ± 0.8 | 97.7 ± 2.6 |
| 1-Octanol | 0.1 | 6.5 ± 2.0 | 28.2 ± 9.8 | 65.3 ± 6.1 |
| | 1 | 6.7 ± 0.3 | 35.7 ± 25.4 | 57.6 ± 8.9 |
| | 10 | 6.7 ± 0.3 | 39.4 ± 15.9 | 53.9 ± 5.5 |
| Abiotic control | | 2.5 ± 0.1 | 14.8 ± 4.8 | 74.9 ± 13.3 |

[a]$^{14}C$-uniformly labeled phenol and $^{14}C$-$C_1$-labeled 1-octanol, in addition to nonlabeled versions of the same compound.
[b]Dosage of nonlabeled compound, in addition to $^{14}C$-labeled compound. For phenol, 5,000 dpm · ml$^{-1}$ (~4 ng · ml$^{-1}$); for 1-octanol, 1,300 dpm · ml$^{-1}$.
[c]Mean values from triplicate incubations ± one standard deviation. Values corrected for total $^{14}C$ recovery.
[d]C in biomass (%): counts of filter after passing solution.
[e]C in $CO_2$ (%): counts of NaOH solution through which gas phase of incubation was passed.
[f]Substrate (%): counts of solution after passing through filter.

assessed for samples at day 3 between the purged gas phase captured in NaOH-solution (indicative of $^{14}C$-$CO_2$), material recovered on a 0.2-$\mu$m filter (indicative of biomass) and 0.2-$\mu$m filtrate (indicative of residual soluble material) (Table 2). $^{14}C$ distributions broadly confirmed the previous inferred mass balances, with between 15.8% (phenol) and 6.5% (1-octanol) $^{14}C$ incorporation into biomass, 28 to 44.5% captured as $^{14}C$-$CO_2$, and the remainder of the $^{14}C$ as nonidentified soluble material (30.7 to 52.4%, depending on the starting concentration) (Table 2). $^{14}C$ carbon incorporation into biomass was slightly lower for 1-octanol than for phenol, but no different for the three starting concentrations (0.1, 1.0, and 10 mg C · liter$^{-1}$; see Table 2). These results indicated, therefore, that substrate biotransformation yields from phenol or 1-octanol into freshwater microbial cell biomass are in the order of 6 to 25%, with 25 to 45% carbon from dosed substrate mineralized as $CO_2$ and under significant formation of nonidentified residual material.

**Community composition analysis.** Even though single substrates were dosed amid a background of undefined AOC, we expected that the substrates might not be metabolized by all community members and would lead to some sort of succession in subgroup changes within the community over time. Such a change in community profiles could be further indicative to some extent for the types of metabolized substrates or their toxicity. In order to discern possible changes in the microbial community composition upon substrate amendment, we deployed both the above-mentioned CellCognize machine learning cell type assignment pipeline for flow cytometry data and 16S rRNA gene amplicon sequence analysis. We restricted 16S rRNA gene amplicon sequencing to incubation endpoints of four of the target compounds and a single dosage of 10 mg C · liter$^{-1}$ (Table S1), all in biological triplicates.

Deconvoluted FCM community data into 32 predefined cell type categories showed subgroup enrichments of benzoate, phenol, and 1-octanol as a function of their dosage between 0.1 and 1,000 mg C · liter$^{-1}$ (Fig. 4A and B). In the case of benzoate, communities followed similar time "paths," which varied more strongly at higher concentrations, before returning to a composition similar as at start (Fig. 4C, benzoate). Community resilience (ability to return to a prior state) and elasticity (time span for the return), as described by Liu et al. (16), clearly increased for the freshwater community

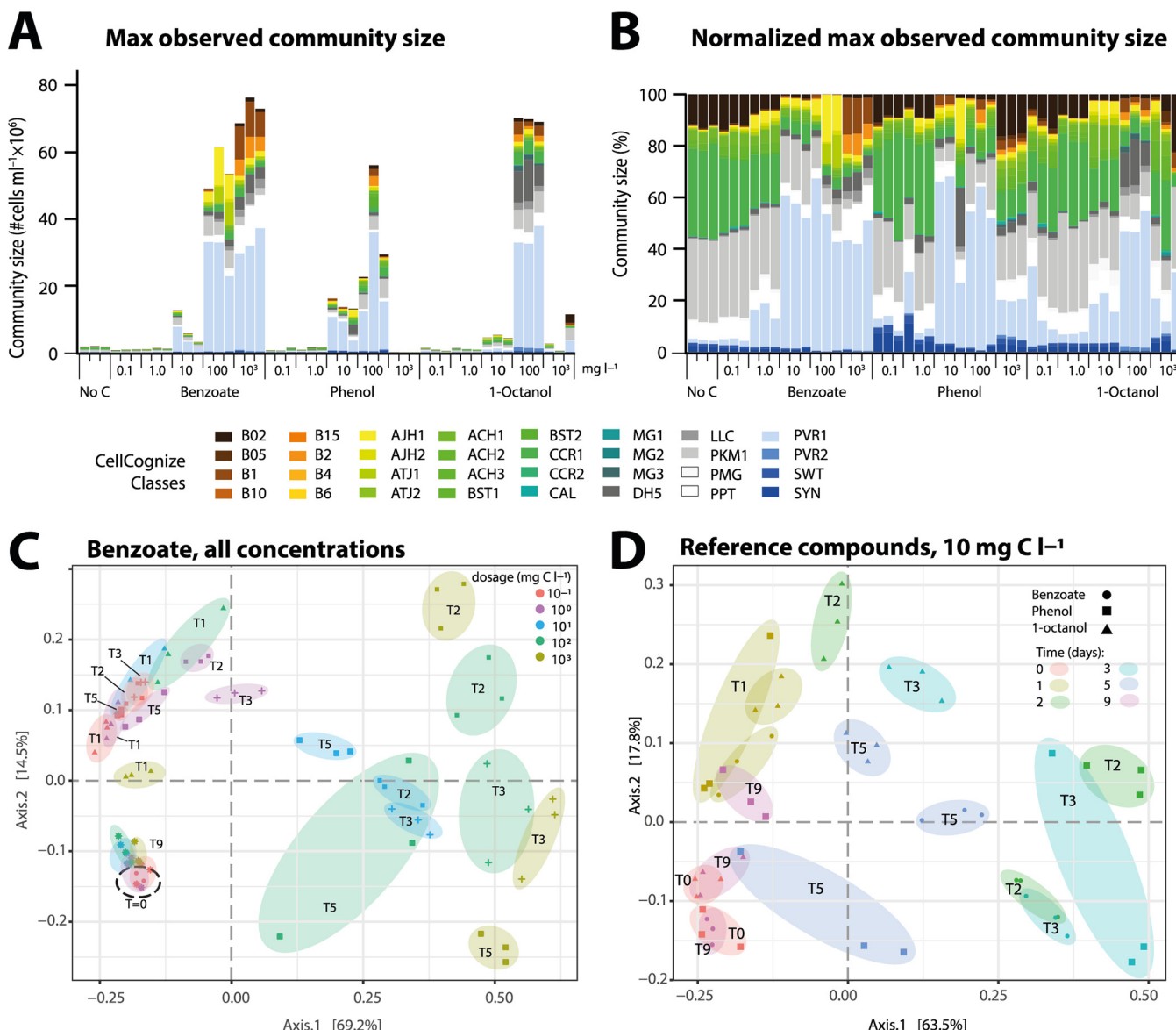

**FIG 4** Community composition changes as a function of compound and starting concentration. (A) Absolute class abundances of attributed cells from each of the community assay replicates in the CellCognize FCM machine learning pipeline. Colors represent class attribution according to the legend in the figure. Stack diagrams produced from community replicate samples at their maximum density, as a function of reference compound and its initial concentration in mg C · liter$^{-1}$ (data from Fig. 1). (B) As in panel A, but shown percent-normalized to the total community size of each individual replicate. (C) Multidimensional scaling (MDS) clustering of CellCognize-deconvoluted community class attributions from incubation assays with benzoate at different starting concentrations, based on Bray-Curtis dissimilarity in the *phyloseq* package in R. Transparently colored ellipsoids group individual replicates per starting concentration with time points (in days) indicated as different symbols and lettering (e.g., T1, 1-day incubation). (D) As in panel C, but for the combined series of benzoate, phenol and 1-octanol at 10 mg C l$^{-1}$ starting concentration. Transparent colored ellipsoids group individual replicates per substrate for the same time point, with substrates indicated by different symbols and time points by lettering as in panel C.

exposed to higher benzoate concentrations, suggesting that the recovery needed a longer time period (see Fig. S3 in the supplemental material).

Time trajectories of community composition development were more variable for phenol and 1-octanol at the different dosages and did not forcibly return to the "starting" position (see Fig. S4 in the supplemental material). It is likely that this was due at least in part to the stronger effect of toxicity at higher dosages (Fig. 1A). Substrate dependency of community development was also evident from incubations at the same nontoxic dosage of 10 mg C · liter$^{-1}$, illustrating the specific enrichment effects of each substrate even at relatively low concentrations (Fig. 4D). Although the extents of community growth were very similar in five independent experiments with lake water

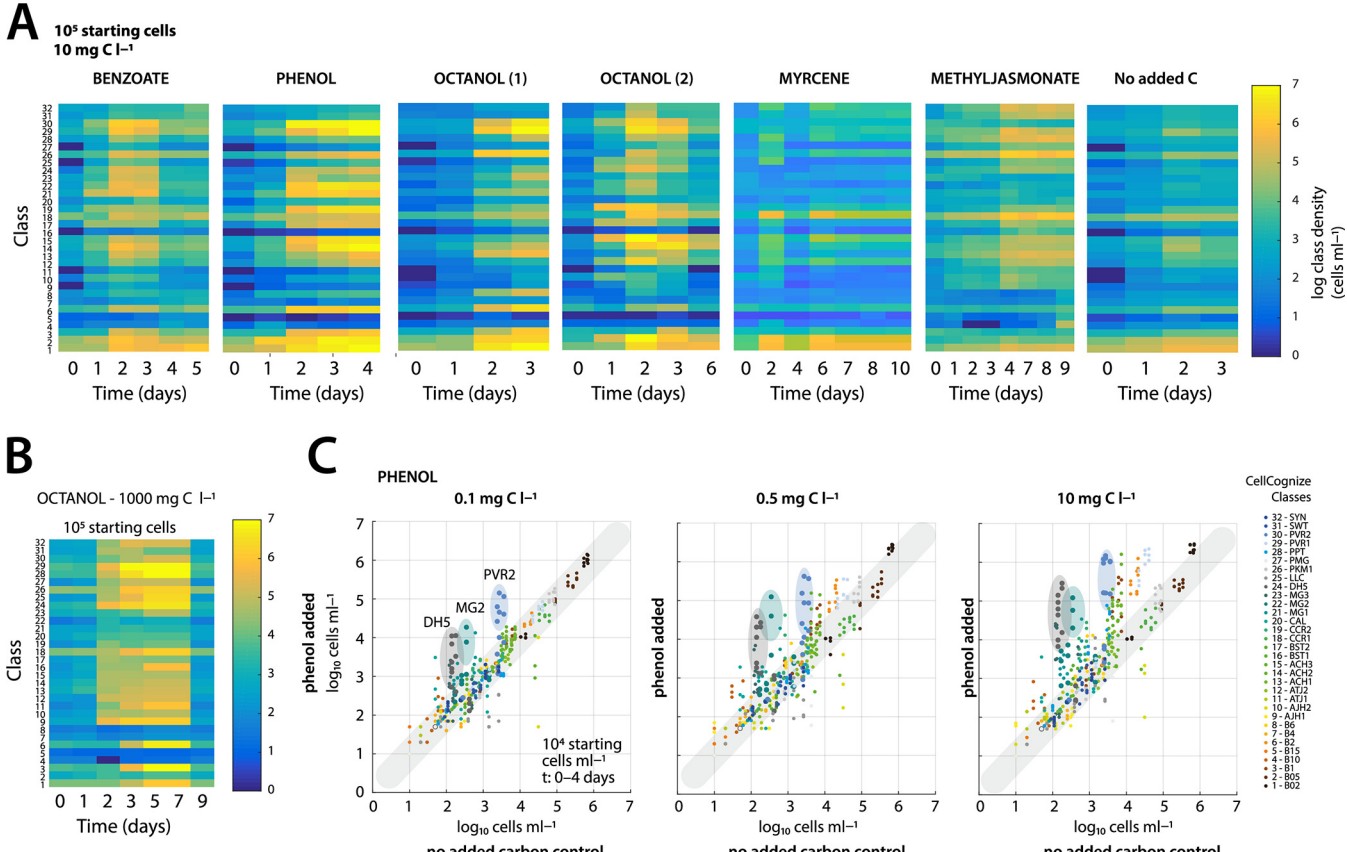

**FIG 5** Subpopulation enrichments in communities dosed with single test compounds. (A) Time changes of mean total cell abundances attributed to the predefined 32 class subgroups deconvoluted from FCM community analysis, plotted on a $\log_{10}$-transformed color heatmap. Individual panels show incubation series at $10^5$ cells $\cdot$ ml$^{-1}$ starting community density per indicated substrate dosed at 10 mg C $\cdot$ liter$^{-1}$. Octanol (1) and octanol (2) denote independent incubation assays. (B) As in panel A, but showing class abundances changes at the toxic inhibitory 1 g C $\cdot$ liter$^{-1}$ 1-octanol concentration (compare to Fig. 1A). (C) Example subpopulation enrichments at three different phenol concentrations in comparison to the parallel incubation assays of the no-carbon control. Dots represent absolute class abundances per replicate (arbitrarily paired between phenol added and no-carbon control series), colored according to the legend on the right. Transparent colored ellipsoids highlight the three selected subpopulations that show consistent and increasing growth as a function of phenol concentration.

microbial community sampled at different seasons (Table S1), the communities did not necessarily develop in the same way (Fig. S1). The trend remained, however, that communities over time "returned" to a similar cell type composition to that at the start (Fig. S1 and S4), suggesting some inherent biological mechanism that controls or restores the temporary blooming of cell types profiting from the newly added carbon substrate.

Amendment of different substrates but at the same concentration (10 mg C $\cdot$ liter$^{-1}$) caused different cell type classes to proliferate (Fig. 5A), indicating that there are substrate type-dependent responses. For most target substrates (except myrcene), multiple cell type classes increased their relative abundances, suggesting that there is not a single principal degrader but several. On the other hand, some succession of cell type class growth could be observed as well. For example, in case of phenol, a rapid response of class 30 (PVR2) occurred, followed by slower increase of classes 12 to 14 (Fig. 5A). The temporary blooming and decline of cell type classes could be seen in longer incubations with benzoate and 1-octanol (e.g., classes 15, 29, and 30; see Fig. 5A). The size increase of specific cell type classes was also dependent on the substrate concentration. For example, the high concentration of 1 g C $\cdot$ liter$^{-1}$ 1-octanol affected multiple classes in comparison to 10 mg C $\cdot$ liter$^{-1}$ (Fig. 5B). Enrichment in three specific classes (class 30, PVR2; class 24, DH5; and class 22, MG2) was observed at the lowest phenol dosage (0.1 mg C $\cdot$ liter$^{-1}$); their cell densities increased at increasing dosages of 0.5 and 10 mg C $\cdot$ liter$^{-1}$, suggesting that they comprise the most active primary

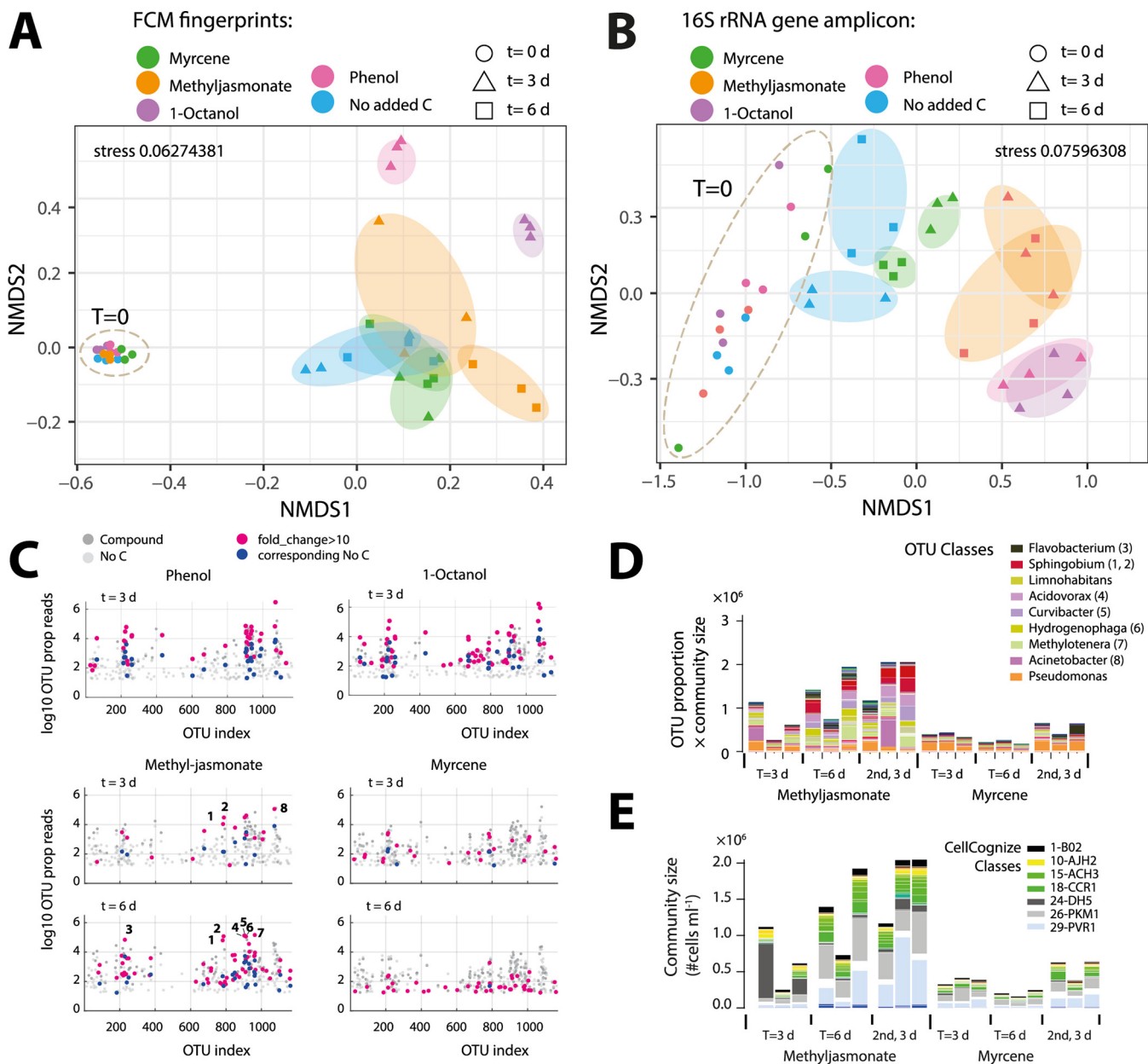

**FIG 6** Community analysis and subpopulation enrichments in assays dosed with phenol, 1-octanol, methyl jasmonate, or myrcene. (A and B) Community composition clustering based on Bray-Curtis dissimilarity distance of methyl jasmonate- and myrcene-incubated series in comparison to 1-octanol and phenol and no-carbon control for the 32-class deconvoluted FCM community (A) or for OTU-attributed classes. (B) 16S rRNA gene amplicon sequence SILVA-classified reads at taxonomy level 3. Stress values are 0.063 (A) and 0.076 (B) for nonmetric multidimensional scaling (NMDS) ordination plots. (C) OTU genus (SILVA-classified reads at taxonomy level 7) enrichments in phenol-, 1-octanol-, methyl jasmonate-, or myrcene-amended communities in comparison to no-carbon controls at the same time point. Dark and light gray dots show mean OTU attributions corrected for the absolute cell abundance from FCM in the respective replicate of the substrate-amended sample or the no-carbon control, respectively. Magenta and dark blue dots denote sample pairs that are >10-fold higher in the substrate-amended sample versus the no-carbon control, respectively (at a false-discovery rate of <0.05 and a $q$ value of <0.05; see Table S3 in the supplemental material for a complete list). Numbers correspond to genus names in panel D. (D) Community size-corrected mean OTU abundances (at SILVA taxonomy level 7) per sample replicate, time. and substrate (i.e., methyl jasmonate and myrcene). (E) As in panel D, but for the 32-class deconvoluted FCM-CellCognize community compositions. Color attributions to the major genera and classes are highlighted in the respective legends. Sample series labeled "2nd, 3 d" represent growth and community composition in a 1:100 diluted transfer from the $t = 6$ primary enrichment on ALW and the same substrate.

phenol degraders in the community at low concentrations (Fig. 5C). In contrast to phenol and 1-octanol, incubations with myrcene and methyl jasmonate led to slower enrichment of specific and different cell type classes (Fig. 5A and Fig. S5 in the supplemental material).

Freshwater communities developed differently over time depending on the substrate, both in the CellCognize classifications (Fig. 6A) and based on 16S rRNA gene

amplicon analysis (Fig. 6B). 16S rRNA gene amplicon analysis suggested some 30 to 50 genera per compound out of 1,168 detected operational taxonomic units (OTUs) to have been statistically significantly enriched upon substrate addition in comparison to incubations of no-carbon controls at the same time point (fold change in community size-corrected proportional read abundance, >10; false-discovery rate, <0.05; $q$ value < 0.05; see Fig. 6C and Table S3 in the supplemental material). Incubations with phenol and 1-octanol became largely dominated by 2 to 4 OTUs and a wider spectrum of less abundant but still 10-fold or more enriched taxa (Table S3), which correspond to the observed enriched cell type classes (Fig. 5C). Some eight OTUs were major representatives in communities dosed with 10 mg C · liter$^{-1}$ of methyl jasmonate, which persisted after a second transfer on the same substrate (Fig. 6D) and may represent the visibly enriched CellCognize classes (Fig. 6E and Fig. S5A). Subgroup enrichments were less pronounced in incubations with 10 mg C · liter$^{-1}$ myrcene (Fig. 6C to E, myrcene), possibly due to the overall poorer community growth on that substrate (Fig. 1D).

The microbial community analyses thus showed that substrate amendments led to distinct substrate- and concentration-dependent community responses, with detectable subpopulation enrichments of likely primary degraders and of those species profiting indirectly from the added carbon.

## DISCUSSION

We show here how flow cytometry can be deployed to simultaneously quantify growth, infer biomass formation, and derive compositional changes of freshwater microbial communities exposed to added chemical substances. Having both kinetic and compositional community information form an important complement to existing ready biodegradability tests that focus on the compound's fate and treat the microbial component as a black box (3). Near-real-time quantification of microbial biomass by FCM and simultaneous distinction of relevant subpopulations within the target microbial community enable a better integrative view of the potential biotransformation of chemical substance input into aquatic ecosystems. Through comparison with classical test methods, we showed that FCM successfully detects biomass formation of known readily biodegradable standard reference (i.e., benzoate, phenol, and 1-octanol) and two fragrance compounds (methyl jasmonate and myrcene), whereas a nonbiodegradable substance (musk xylene) remained without effect. We tested a range of compound dosages between 0.1 mg C and 1 g C · liter$^{-1}$ and the effects of different community starting cell densities and of seasonality, showing that concentrations of 1 mg C · liter$^{-1}$ and $10^5$ starting cells · ml$^{-1}$ are sufficient to observe specific community growth. The lowest direct compound-specific induced community growth was observed for 0.1 mg C · liter$^{-1}$ phenol or 1-octanol, which is in the range of what has been achieved in terms of community growth on natural AOC (19, 20). Furthermore, independent assay repetitions prepared with freshwater inoculum from the same site at different times of the year showed very consistent results, indicating limited variability of the test results depending on the inoculum seasonality.

In order to properly describe compound biodegradability, one needs to be able to calculate the net biomass formation at the expense of the added compound. In order to do this, we transformed community growth (in number of cells · ml$^{-1}$) into biomass (as g C · liter$^{-1}$). This has been previously done using a global single mean individual cell mass of 100 fg C (26), which we refined here by deploying the CellCognize FCM classifier algorithm that assigns cells into predefined standard classes with defined mass. The net community biomass thus calculated from the increase in different cell types was in good agreement with alternative yield estimates from $^{14}$C-substrate incubations. For phenol and 1-octanol, biomass incorporation reached 15 to 30%, which is in good agreement with literature data (27, 28). In contrast, the proportion of carbon released as $CO_2$ from both phenol and 1-octanol, at three different concentrations (0.1 to 10 mg C · liter$^{-1}$), was lower than expected from complete mineralization (~60%) (29), with a higher-than-expected formation of soluble transformation product (29).

This may be specific for the freshwater microbial community transformation used here, but such an outcome would result in a "failure" for a ready biodegradation test (3), whereas our data indicate complete growth.

For biodegradable fragrance compounds (i.e., methyl jasmonate and myrcene) dosed at 10 mg C · l$^{-1}$, we deduced biomass incorporation between 2.6 and 6.6%, whereas $CO_2$ evolution for methyl jasmonate indicated similar mineralization (~27%) as for phenol and 1-octanol. The smaller percentage of carbon to biomass conversion for methyl jasmonate, $CO_2$ evolution, and the complete disappearance of the parent would indicate partial utilization of the compound for microbial growth. The lower proportion of $CO_2$ evolved from myrcene but similar biomass incorporation as methyl jasmonate may have resulted from its low water solubility and relatively high vapor constant (see Table S4 in the supplemental material), causing poor compound bioavailability to the microbiota.

The inoculum and cell density are considered to be among the critical aspects of any biodegradation test (3, 5, 9). Not only is the absolute species diversity in the inoculum determining for its potential to contain the capacity to carry out the requested biotransformation reaction (10), but the initial concentration of the "degrader" species limits the observed transformation kinetics and thus the probability for pass/failure of the test within the assay duration (7). Inoculum densities may be increased artificially for the test to increase the likelihood of a true-positive outcome (9), but this is debated in terms of its ecological significance (5). We chose here to keep starting cell densities similar to those in their natural freshwater aquatic environment (10$^4$ to 10$^6$ cells · ml$^{-1}$), but this has its own challenges in terms of our FCM-based methodology. For example, higher starting cell densities of 10$^6$ cells · ml$^{-1}$ would have a higher probability to contain the targeted degrader bacteria, but at low added carbon dosages (e.g., 0.1 to 1.0 mg C · liter$^{-1}$) may yield too few net new cells to be reliably quantified. Indeed, we find that kinetics of growth were less pronounced at high than at low starting cell densities. In contrast, we expected that low cell densities (here, 10$^4$ cells · ml$^{-1}$) might limit starting microbial diversity and decrease the chance of having sampled microorganisms in the assay capable of carrying out the biodegradation of the added chemical. At least for the three reference compounds used here, however, we observed rapidly biodegradation independent of the starting cell concentration (except at the highest dosage of 1 g C · liter$^{-1}$), although longer lag phases were observed for initial cell densities of 10$^4$ cells · ml$^{-1}$. Incubations with the two biodegradable fragrances, in contrast, showed that a low substrate dosage may not lead to detectable community growth.

As community growth is judged in comparison to a no-carbon control, the presence of background assimilable organic carbon in the medium is critical for the performance of the methodology, in two ways. As was pointed out previously (23), external substrate dosages in this type of assay and these concentrations should be considered mixed-carbon cultures, and degrader species may be carbon generalists that profit from different simultaneously available carbon substrates instead of specialists that would be dependent only on the added substrate. On the other hand, the estimated 60 to 100 $\mu$g · liter$^{-1}$ AOC in the ALW medium on its own permits 3 or 4 generations of community growth, and this influences the reliably detected community increase as a consequence of the added substrate. Striving for even purer test water may be counterproductive, because this may limit many generalist bacteria to sustain. One way forward to become more accurate even at lower cell densities is to unravel the community into subgroups, as we show here by the use of the FCM classifier algorithm, because this may indicate specific subgroup enrichments as low as 10$^4$ cells · ml$^{-1}$ amid a total community density 100 times as large.

The community compositional data in the substrate incubations clearly indicated compound-specific changes, which were concentration dependent. Furthermore, all substrates led to enrichment of multiple subgroups (e.g., detectable from CellCognize) or genera (from 16S rRNA gene amplicon sequencing). This suggests that at least for the compounds tested here, there was not a single specialist degrader bacterium

present in the inoculum, but several that may have contributed to the utilization of the added carbon source by the community. The fact that community compositional changes were compound specific further indicates that different bacterial groups contribute to the capacity of freshwater communities to deal with a variety of chemical inputs. Many of the CellCognize classes enriched on benzoate, phenol, and 1-octanol were represented by standards of *Pseudomonas* (e.g., PVR1; *Pseudomonas veronii*, PPT; *Pseudomonas putida*) or *Acinetobacter* genera (e.g., AJH1; *Acinetobacter johnsonii*, ATJ1; *Acinetobacter tjernbergiae*), which have been frequently implicated in degradation of aromatic and aliphatic compounds (30). The CellCognize community composition deconvolution technique is particularly promising, as it gave almost instant additional information from FCM data concerning potential subgroup enrichments.

In conclusion, we benchmarked and expanded the usefulness of deploying FCM-based community growth studies to study and infer compound biodegradability at relatively low concentrations using native and relevant microbial communities. Given the rapidity and ease of the method, and low (100- to 200-$\mu$l) sample volume requirements, FCM can be quickly scaled up to test many compounds and concentrations simultaneously. As the system is flexible, various other communities (e.g., derived from activated sludge or top soils) may be tested for their capacity to degrade the tester compounds. In that case, cells are to be detached from particles before FCM measurement in order to be properly quantified. Finally, more attention needs to be given to further refine the deconvolution of microbial communities into meaningful classes. Our data and those of others (14, 15, 17, 18, 31–33) have shown that rough trends and broad cell type classes can be distinguished, even in complex community cell mixtures, by using both supervised and unsupervised machine learning methods. Since these were mostly proof-of-principle studies, we expect that this field of cell recognition will advance further and become more accurate in classifying cell types to taxonomic relevance while simultaneously recognizing physiologically relevant parameters (dividing or damaged cells). This would not only be helpful for biodegradability tests but crucial for many quantitatively focused microbiome studies.

## MATERIALS AND METHODS

**Flow cytometry analysis.** Community cell numbers were enumerated by flow cytometry total cell counting (13) in suspensions of between $10^4$ and $10^7$ cells $\cdot$ ml$^{-1}$. Microbial cells in aqueous samples (volume of 200 $\mu$l) were stained with 2 $\mu$l SYBR green I solution (10 $\mu$l $\cdot$ ml$^{-1}$ in dimethyl sulfoxide [DMSO]; Molecular Probes), and incubated in a 96-flat-bottomed-well plate in the dark for 15 min at room temperature. To count dead and compromised cells, suspensions were additionally stained with 2 $\mu$l propidium iodide (PI) solution (10 $\mu$g $\cdot$ ml$^{-1}$; Molecular Probes). Samples with cell counts above $10^7$ per ml were diluted with phosphate-buffered saline (PBS). Stained samples (20 $\mu$l) were aspirated on a NovoCyte flow cytometer (ACEA Biosciences, Inc.) at a flow rate of 14 $\mu$l $\cdot$ min$^{-1}$ with a maximum acquisition rate of 35,000 events $\cdot$ s$^{-1}$. The NovoCyte has accurate volumetric-based cell counting hardware, and no calibration through addition of external beads is necessary. The sheath flow rate was fixed at 6.5 ml $\cdot$ min$^{-1}$, which corresponds to a core diameter of approximately 7.7 $\mu$m. SYBR green I fluorescence was excited at 497 nm and collected at 520 $\pm$ 30 nm (fluorescein isothiocyanate [FITC] channel), whereas PI was excited at 535 nm and its fluorescence was collected at 617 $\pm$ 30 nm (PI channel). Sideward (SSC) and forward scattered (FSC) light intensities were also recorded. Ultrapure water samples (in-house Milli-Q) were analyzed to determine particle background levels and set electronic threshold on the instrument, below which particles were considered to be abiotic or to originate from cell debris. This threshold resulted in a value of 600 in the FITC-H channel and of 20 in the FSC-H channel. Instrument threshold settings and electronic gates were kept the same for all samples in all experiments in order to obtain comparable data. Raw FCM output values (FCS-H/-A, SSC-H/-A, FITC-H/-A, and width) were exported as .csv files, which served as input for the CellCognize machine learning pipeline for cell type classification (see below and in reference 15). This process retains values within the set parameter thresholds and removes events with negative values in any of the FCM output values. Events passing through this pipeline were counted. and their sum is reported as the community size determination (in cells $\cdot$ ml$^{-1}$).

**Test compounds and dosing.** The chemical compounds tested in this study and their main properties are listed in Table S4 in the supplemental material. Sodium benzoate, phenol, and 1-octanol were purchased from Sigma-Aldrich at a purity of more than 95%. Methyl jasmonate, musk xylene, and myrcene were provided by Firmenich SA, Geneva, Switzerland. To avoid introducing additional carbon from solvents, we dosed all test chemicals either directly from the pure compound to the assays or from a 10- to 100-fold concentrated aqueous solution in artificial lake water medium (ALW; see Table S2 in the supplemental material). Direct dosing of pure chemicals in liquid form (i.e., methyl jasmonate, myrcene, and 1-octanol) to the test bottles was achieved with an ultra-accurate glass displacement microinjection

pipet (Drummond Scientific, USA) delivering volumes as low as 30 nl. Musk xylene was dosed directly into the test bottles after being weighed on an analytical balance. Benzoate, phenol, and 1-octanol were prepared in aqueous stock solutions of up to 1 g C $\cdot$ liter$^{-1}$ and then volumetrically diluted to the final initial compound concentration (depending on the experiment, between 0.1 mg C $\cdot$ liter$^{-1}$ and 1 g C $\cdot$ liter$^{-1}$) in the test bottles.

**Preparation of microbial cell suspension.** Community cell suspension was freshly prepared from collected Lake Geneva water by filtration and resuspension. Approximately 10 liters of freshwater were sampled at 10 m from the shore near St. Sulpice (Switzerland) by lowering and filling a Teflon carboy below the surface, which was transported immediately (within 20 min) to the laboratory. Debris were removed by filtering the sample entirely through a 40-$\mu$m-pore-size nylon cell strainer (Falcon, USA). Microbial cells were then collected from the filtrate by passage through a 0.2-$\mu$m-pore-size polyethersulfone membrane filter (Sartorius, Switzerland). The filter with cells was subsequently placed in 100 ml of ALW for at least 2 h to release and resuspend the cells, after which the suspension was serially diluted and stained, and the cell concentrations were quantified by FCM. Depending on the intended experiment, the inoculum was then diluted to achieve starting cell densities of approximately 10$^4$, 10$^5$, or 10$^6$ cells $\cdot$ ml$^{-1}$.

**Medium for community growth experiments.** Community growth experiments were carried out in ALW medium, which has a composition similar to that of Lake Geneva water (Table S2). ALW was prepared by dissolving, per liter of ultrapure water, 36.4 mg CaCl$_2\cdot$2H$_2$O, 0.25 mg FeCl$_3\cdot$6H$_2$O, 112.5 mg MgSO$_4\cdot$7H$_2$O, 43.5 mg K$_2$HPO$_4$, 17 mg KH$_2$PO$_4$, 33.4 mg Na$_2$HPO$_4\cdot$2H$_2$O, and 25 mg NH$_4$NO$_3$. ALW medium was sterilized by filtering through a 0.2-$\mu$m-pore-size polyethersulfone membrane into a 2-liter volume acid-washed glass bottle with automated dispenser.

**Community growth assays.** Growth assays were conducted in triplicates for each treatment or condition, with 100 ml of ALW medium in acid-washed and autoclaved 500-ml Schott flasks, which was amended with one of the test chemicals (see Table S1 in the supplemental material for an overview of all test series). The freshly prepared microbial community inoculum was added last. Flasks were then tightly closed with Teflon-lined septa and screw caps, mixed, sampled again for a time zero measurement, and then incubated in the dark at 21°C and with 150 rpm rotary shaking. Assays were sampled at regular time intervals, typically once per day for a period of up to 10 days. Aqueous phase samples were taken by suction from a long metal needle attached to a syringe inserted through the cap septum, to avoid opening the caps and losing volatile substrate (note that this is a compromise, and some substrate loss may occur through the tiny hole punched in the Teflon-lined septum). For all assays, triplicate "no-carbon" controls without any added carbon were inoculated in parallel in order to estimate background community growth on the AOC of the medium itself.

In order to test the effect of dead cells on community growth, we pasteurized a 2-ml aliquot of the freshwater suspension in an Eppendorf tube for 15 min at 75°C under vigorous shaking (600 rpm). After this treatment, 80% of cells stained positive with propidium iodide and low with SYBR green I, indicative for intact but compromised cells. Only 2% of the cells in the initial freshwater suspension fell into the same gate (high propidium iodide and low SYBR green I signal). Triplicate assays were then started in ALW with 10$^6$ cells $\cdot$ ml$^{-1}$, in the presence or absence of phenol at 10 mg C $\cdot$ liter$^{-1}$, and containing 25% or not of dead cells. Assays were incubated, sampled, and measured by FCM as described above.

**Estimation of kinetic and stoichiometric parameters.** The apparent community maximum specific growth rate was estimated from measured community cell numbers as a function of incubation time on the different substrates using the *flexible modeling environment* (FME) package in R (version 1.3.6.1). Measured data were numerically and simultaneously solved for growth kinetics by minimizing the sum of the squared residuals between measured and modeled data. We refer to the time points with the highest measured cell numbers as community stationary phase.

**Cell type categorization and biomass estimation.** Microbial cells in community samples were classified according to their multidimensional resemblance to a set of 32 predefined cell type standards, among which were eight types of microbeads with different diameters, 23 bacterial strains and physiological subtypes, and one yeast strain. A machine learning artificial neural network had been previously trained to recognize and differentiate individual cells comprising the 32 standards (15). The resulting classifier algorithms were used here to calculate the probability of resemblance of the seven recorded FCM parameter values for individual unknown cells to each of the categories, and subsequently to attribute that cell to the category with the highest resemblance. FCM .csv exported output files were used as input for the classifier algorithm as described in detail with examples in reference 34, and each particle event passing the thresholds for each of the seven FCM parameters was quantitatively deconvoluted into one of the 32 cell types by multiparametric resemblance. This classification was then further used either for fingerprinting analyses or to quantify growth of subgroups (as encompassed by cell types) compared to that in the no-carbon controls.

**16S rRNA gene amplicon sequencing.** Freshwater communities from a set of triplicate assay series (incubations with phenol, 1-octanol, methyl jasmonate, or myrcene, each at a dosage of 10 mg C $\cdot$ liter$^{-1}$, and no-carbon control) were characterized by 16S rRNA gene amplicon sequencing and simultaneously measured by FCM. The data of incubations with phenol and 1-octanol were retrieved from BioProject accession number PRJNA641590 (15). Samples for amplicon sequencing at $t = 3$ and $t = 6$ days were volume adjusted based on their FCM cell counts to the $t = 0$ sample in order to collect similar amounts of cells on 0.2-$\mu$m membrane filters (PES, Sartorius) for DNA extraction. Cells were stored on filters in FastDNA Spin kit solution for soil (MPBio) at −80°C until analysis. DNA was extracted by using a FastDNA Spin kit for soil according to the manufacturer's instructions (MPBio). The V3-V4 hypervariable region of the 16S rRNA gene was amplified using the 341f/785r primer set with appropriate Illumina adapters and

barcodes, following recommendations for PCR conditions, amplifications and library preparations described in the Illumina Amplicon sequencing protocol (https://support.illumina.com/documents/documentation/chemistry_documentation/16s/16s-metagenomic-library-prep-guide-15044223-b.pdf). Equal amounts of amplified DNA from each sample were pooled and sequenced bidirectionally on the Illumina MiSeq platform at the University of Lausanne Genome Technologies Facility. Raw 16S rRNA gene amplicon sequences were quality filtered, concatenated, verified for absence of potential chimeras, dereplicated, and mapped to known bacterial taxonomy using QIIME 2 at 99% similarity to the SILVA taxonomic reference gene database on a UNIX platform (35). SILVA operationally defined taxonomic units (OTUs) with fewer than 50 reads per sample (from a total of approximately $10^5$ reads per sample) were removed from further analysis, after which OTU counts were corrected for the absolute FCM cell count in the respective sample or normalized to the same total read number across all samples (expressed as percentages).

**Community fingerprinting.** The community composition in the different experimental series was characterized from the percent-normalized or the FCM total community size corrected OTU read numbers, and further from the 32-subgroup FCM classification procedure.

Compositional similarities between samples were calculated using the Bray-Curtis dissimilarity index from relative abundance of each taxon (at level 3, to have the same number of groups as in FCM) or FCM fingerprints (i.e., summing over all 32 classes per sample results in a value of 100) and visualized on multidimensional scaling plots using the *phyloseq* package in R.

Community profile heatmaps of the 32 subgroup classifications in sample series were produced in MatLab by calculating and plotting the mean of cell counts $\cdot$ ml$^{-1}$ attributed to each of the 32 subgroups across biological triplicates on a standardized $\log_{10}$-color scale for all samples (0 to 7). Significant enrichments were visualized in scatterplots of $\log_{10}$ subgroup counts of all replicates and time points of substrate-incubated series (e.g., phenol at 0.1 mg C $\cdot$ liter$^{-1}$) versus the parallel no-carbon control, with the null assumption being that there is no difference at any time point from the no-carbon control (i.e., gray zones plotted in Fig. 5B). Significance of OTU enrichment (i.e., at SILVA taxonomy level 7) was tested in a substrate-incubated sample versus the no-carbon control at the same incubation time point comparison of triplicate community size-corrected OTU counts, using the *mattest* function (MatLab 2016a) for gene expression differences, under 1,000 permutations, with a fold change cutoff 10, a false-discovery rate of $< 0.05$ and a $q$ value of $< 0.05$ (see Table S3 in the supplemental material).

**Mass balance experiments.** Three parallel separate incubation series of biological triplicates with all compounds except benzoate, against no-carbon control and an abiotic control, were used to determine the carbon mass balance of dosed substrate at 10 mg C $\cdot$ liter$^{-1}$ into community biomass. One series of triplicates was used to quantify community growth by FCM. The second series consisted of triplicate flasks for each time point (e.g., 0, 2, 24, 48, 72, 144, and 168 h), which were extracted three times by dichloromethane to measure loss of the parent compound by gas chromatography on an Agilent system 7890 gas chromatograph equipped with two Agilent DB-1MS columns (60 m $\times$ 0.25 mm $\times$ 0.25 $\mu$m, part number 122.0162) connected to two detectors, a 5975 mass spectrometer and a flame ionization detector. Samples (1 $\mu$l) were injected at 250°C in split mode (1:50). The initial oven temperature was maintained at 60°C for 1 min, increased to 240°C at 5°C $\cdot$ min$^{-1}$, then increased to 300°C at 10°C $\cdot$ min$^{-1}$ and held for 1 min. Mass spectra were generated at 70 eV with an $m/z$ 29 to 250 U mass range from 0 to 20 min, followed by 29 to 450 U. Linear retention indices (LRI) were determined from the injection of a series of $n$-alkanes (C$_5$ to C$_{31}$) under identical conditions. GC-MS peaks were identified and integrated using HP-Chemstation software and internal MS and LRI libraries. Identification was performed by gas chromatography (GC)-mass spectrometry (MS), and quantitative analysis was performed by GC-flame ionization detection (FID) using methyl octanoate as an internal standard. Relative response factors were calculated for each compound.

The third series of triplicates was used to measure $CO_2$ evolution over 24 days in a test according to Birch and Fletcher (36), equivalent to that described in reference 6, with two blank controls and three Erlenmeyer flasks with test compound. $CO_2$ concentrations of both liquid and gas phase were determined as described by Birch and Fletcher (36).

Cell type attributions from FCM data (see above) were used to estimate biomass growth from community cell counts. For this, we referred to the previously estimated cell volume and mass of each of the 32 standards (15) and assumed that cells from unknown samples attributed to a standard category would be similar in their per cell mass. Cell counts per subgroup were thus multiplied by the per cell subgroup mass and then summed across all 32 subgroups to obtain the community biomass.

**$^{14}$C mass balance.** To further corroborate the mass balance, we repeated incubations in biological triplicates on two separate occasions with two substrates for which $^{14}$C-labeled compound was commercially available (phenol and 1-octanol) against no-carbon control (to determine background growth) and abiotic controls (to determine $^{14}$C label losses in the procedure). Uniformly $^{14}$C-labeled phenol (at 5,000 dpm $\cdot$ ml$^{-1}$) or C$_1$-labeled $^{14}$C–1-octanol (Anava Trading SA) (at 1,300 dpm $\cdot$ ml$^{-1}$) were dosed to each flask and in mixture with unlabeled compound of the same at either 0.1, 1.0 or 10 mg C $\cdot$ liter$^{-1}$. The flasks were incubated at 21°C in the dark with 150 rpm rotary shaking for 3 days. Immediate and daily sampling served to follow community growth by FCM. At apparent community stationary phase (day 3), 12 ml was sampled from the aqueous solution of each flask for $^{14}$C analysis, without opening the caps. A subsample of 0.1 ml was taken for measuring the radioactivity in the aqueous solution. A 5-ml aliquot was filtered through 0.2-$\mu$m-pore-size membrane filter to collect cell biomass, and a comparison subsample (0.1 ml) was taken from the filtrate. The remaining solution after sampling (85 ml) was acidified to pH 3, and $CO_2$ was purged from the liquid by air stripping during 1 h and collected into three subsequent vials, each with 5 ml of 1 M NaOH. Vials were pooled, and 0.5 ml was sampled for $^{14}$C counting.

Samples or filters were mixed in 5 ml liquid scintillation cocktail (Perkin Elmer) to measure the amount of $^{14}C$ cpm, which was converted to dpm by a factor of 1/0.94 to correct for the instrument's efficiency. $^{14}C$ counts in substrate-amended samples were corrected for the total recovery. $^{14}C$ biomass and $^{14}C$ $CO_2$ values were corrected for the $^{14}C$ counts in the corresponding fractions of the abiotic controls. Similar mass partitioning was assumed for the unlabeled compound fraction to calculate the net community biomass formation in percentage of the added substrate carbon weight (Table 2).

**Data availability.** Raw sequence reads from 16S rRNA V3-V4 amplicon sequencing of the various samples described here are available from the Sequence Read Archive under BioProject accession number PRJNA641590. Raw flow cytometry data from all samples analyzed and processed in this study can be accessed from Flow Repository under accession entry FR-FCM-Z332.

## SUPPLEMENTAL MATERIAL

Supplemental material is available online only.
**FIG S1**, PDF file, 0.5 MB.
**FIG S2**, PDF file, 0.4 MB.
**FIG S3**, PDF file, 0.4 MB.
**FIG S4**, PDF file, 0.1 MB.
**FIG S5**, PDF file, 0.6 MB.
**TABLE S1**, DOCX file, 0.1 MB.
**TABLE S2**, DOCX file, 0.1 MB.
**TABLE S3**, DOCX file, 0.1 MB.
**TABLE S4**, DOCX file, 0.1 MB.

## ACKNOWLEDGMENTS

We thank Siham Beggah Möller for her initial help in lake water community counting by flow cytometry.

This work was supported by grant 16800.1 PFIW-IW from the Swiss Commission for Technology and Innovation.

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
