## [Reviewer comments · mSystems]

Assessing biodegradability of chemical compounds from microbial community growth using flow cytometry

Birge Özel-Duygan, Sylvain Rey, Sabine Leocata, Lucie Baroux, Markus Seyfried, and Jan van der Meer

Corresponding Author(s): Jan van der Meer, University of Lausanne

Review Timeline:

Submission Date:	October 30, 2020
Editorial Decision:	December 14, 2020
Revision Received:	January 15, 2021
Accepted:	January 15, 2021

Editor: Laura Hug

Reviewer(s): Disclosure of reviewer identity is with reference to reviewer comments included in decision letter(s). The following individuals involved in review of your submission have agreed to reveal their identity: Jim Spain (Reviewer #1)

Transaction Report:

DOI: <https://doi.org/10.1128/mSystems.01143-20>

December 14, 2020

Prof. Jan Roelof van der Meer
University of Lausanne
Department of Fundamental Microbiology
Batiment Biophore
Quartier Unil-Sorge
Lausanne CH 1015
Switzerland

Re: mSystems01143-20 (Assessing biodegradability of chemical compounds from microbial community growth using flow cytometry)

Dear Prof. Jan Roelof van der Meer:

Your paper was reviewed by two expert reviewers who both expressed that the work was interesting, scientifically sound, and relevant to the scope of MSystems. Each identified suggested revisions and areas lacking clarity that need to be addressed before the paper can be published. I look forward to seeing your revised manuscript.

Below you will find the comments of the reviewers.

To submit your modified manuscript, log onto the eJP submission site at <https://msystems.msubmit.net/cgi-bin/main.plex>. If you cannot remember your password, click the "Can't remember your password?" link and follow the instructions on the screen. Go to Author Tasks and click the appropriate manuscript title to begin the resubmission process. The information that you entered when you first submitted the paper will be displayed. Please update the information as necessary. Provide (1) point-by-point responses to the issues raised by the reviewers as file type "Response to Reviewers," not in your cover letter, and (2) a PDF file that indicates the changes from the original submission (by highlighting or underlining the changes) as file type "Marked Up Manuscript - For Review Only."

Due to the SARS-CoV-2 pandemic, our typical 60 day deadline for revisions will not be applied. I hope that you will be able to submit a revised manuscript soon, but want to reassure you that the journal will be flexible in terms of timing, particularly if experimental revisions are needed. When you are ready to resubmit, please know that our staff and Editors are working remotely and handling submissions without delay. If you do not wish to modify the manuscript and prefer to submit it to another journal, please notify me of your decision immediately so that the manuscript may be formally withdrawn from consideration by mSystems.

Corresponding authors may join or renew ASM membership to obtain discounts on publication fees. Need to upgrade your membership level? Please contact Customer Service at

Service@asmusa.org.

Sincerely,

Laura Hug

Editor, mSystems

Journals Department
Reviewer comments:

Reviewer #1 (Comments for the Author):

MSystems 1143-20 describes a systems approach to evaluation of microbial responses to chemical pollutants using a combination of cutting- edge technologies along with more traditional isotopic tracer analyses. The approach is a logical extension of the previous work described in reference 18 and elsewhere. The work seems technically sound and the presentation is clear. A particular strength is the consideration of growth rather than excessive reliance on omics. Minor suggestions for improving the presentation are listed below.

Specific comments:

1. The criticisms of current standard approaches to biodegradability testing could be dialed back a bit. Or provide examples where the standard approaches have led to bad regulatory decisions that would have been reversed by the approach described here.
2. The broad applications for microbial ecology could be emphasized more with biodegradability testing for regulatory decisions as one example. It wouldn't hurt to mention the demands of REACH, with the caveat that the strategy described here might not be amenable to high throughput.
3. Writing is clear, but a few passages should be reworded for simplicity and clarity. Examples include but not limited to: Lines 15-19, 316-319. Minimize "environmental compartment", "benchmarking", "framework", "methodological" and combinations thereof.
4. Line 21. Various
5. Line 48. Faster than what? Why is faster important?
6. Lines 73-76. Tell the reader why it would be important and how it would change regulatory decisions.
7. Where possible minimize sections in the Results that read the data.
8. Lines 311-314, 396-398. Tell the reader what is new.
9. Lines 331-332. Provide reference to support the statement.
10. Line 408. "...we feel relatively confident..." Replace with confidence intervals or something similar.
11. Lines 499-511. Briefly explain how changes in cell size/shape during the growth cycle were accommodated by the system.
12. Line 439. Likely benzoic acid or sodium benzoate was purchased.

13. Lines 405-411. Tell the readers about the strengths and weaknesses, barriers to practical application, questions remaining, quantitative comparison with standard methods, potential applications in other areas of microbial ecology. For example, discuss potential problems of cell sorting with sludge/soil, loss of resolution as members of specific populations undergo dramatic changes in cell size and shape during the growth cycle, limitations of amplicon sequencing for following populations of specific degraders.

Reviewer #2 (Comments for the Author):

The work presented is a novel approach to understanding biodegradation of organics by identifying cell types using a technique developed by the authors (the CellCognize pipeline) to provide a higher resolution in identifying microbial community changes. Data and the approach summarized could be highly impactful in identifying populations in mixed communities that are impacted by xenobiotics and the interrelationship to transformation of organic substrates. The authors used phenol, 1-octanol, and benzoate as the reference compounds and compared this to three others with two compounds with intermediate biodegradability (in addition to a non-biodegradable control). The authors determined growth rates of a freshwater microbial community when using the reference substrates as well as the incorporation of C14-labelled carbon into the biomass and evolution to carbon dioxide. Subgroups that were enriched by substrates were identified based on the CellCognize pipeline and the 16S amplicon data. Using flow cytometry, this approach will substantially change how we understand responses of complex bacterial communities to substrates. The authors should organization of the text and components in the figures that will make it easier for the reader to connect statements and the data presented in figures/tables. There are also a few discrepancies to be clarified in terms of the experimental approach and analysis as outlined in the specific comments below.

General Comments:

1. Is there a reason to use up to 100 mg/L of the specific substrates? Such as, how environmentally relevant would these concentrations be to understanding impacts within freshwater systems.
2. Figures, methods, and results some times have a different order of the substrates presented. Please check that there is consistency in the text and in the figures/tables being referred to as this presents some confusion when visually comparing the statements and figures.
3. The numbering for the supplementary figures are presented in a different order than mentioned in the text.

Abstract

29: 'significant community growth'? by what statistical analysis was this significant?

31: 'notable enrichment subgroups visible already at 0.1 mg C/l initial compound concentration' Is this for all three compounds? Also, hat about benzoate? There is no mention in the abstract on the results for this compound.

Results:

150-161: Community growth rates increased at up to 10 mg C/L but in 1-octanol, the 100 mg C/L concentration had a larger community size than the 1,000 mg C/L treatment (Fig. 1A). This is not mentioned in the text. Is there an explanation for this? Is it a function of the exposure to octanol or an artifact of the study approach?

187: 'on average 25% more cells' Check this again. Some comparisons are less than 10% different.

190: 'average 10% less cells at the same sampling point' t=2 comparison in the figure shows similar

numbers and the t=3 comparison is much less than 10%. Please check this.

205: 'between 22-32% for both compounds' Figures 3 does not show this. Please check these numbers.

225: It is mentioned that the 50% of phenol was not recovered, but the table shows 51.6%.

260: It's mentioned that CellCognize cell types and the 16S rRNA gene amplicon sequence data was used for the community composition analysis. Was this for all substrates? The Methods mentioned 'notable methyl jasmonate and myrcene'. Was the 16S analysis performed on the other substrates but not presented in the manuscript?

284-285: How can this assumption be made using cell type class only and not 16S data? It is unclear if 16S data was used.

298: What about phenol and 1-octanol in terms of this data? Was this not performed? Any specific OTUs?

Methods:

454: Were samples kept on ice during transport? How were water samples stored between field site and lab?

474: Should 'at last' be 'last'?

512: Why not the phenol or 1-octanol treatments?

556: Why not benzoate?

Table 2: Should format as presented in Table 1

Figure 3b: Scale should be changed to show 25% Co₂ as the maximum point just like the other figures.

Figure S1: What does InitialCond and HighConc in the legend mean?

Figure S1, S3 captions should immediately identify that the community data is based on cell type classifications.

Review mSystems01143-20: Assessing biodegradability of chemical compounds from microbial community growth using flow cytometry

Comments and Suggestions for the Author:

The work presented is a novel approach to understanding biodegradation of organics by identifying cell types using a technique developed by the authors (the CellCognize pipeline) to provide a higher resolution in identifying microbial community changes. Data and the approach summarized could be highly impactful in identifying populations in mixed communities that are impacted by xenobiotics and the interrelationship to transformation of organic substrates. The authors used phenol, 1-octanol, and benzoate as the reference compounds and compared this to three others with two compounds with intermediate biodegradability (in addition to a non-biodegradable control). The authors determined growth rates of a freshwater microbial community when using the reference substrates as well as the incorporation of C14-labelled carbon into the biomass and evolution to carbon dioxide. Subgroups that were enriched by substrates were identified based on the CellCognize pipeline and the 16S amplicon data. Using flow cytometry, this approach will substantially change how we understand responses of complex bacterial communities to substrates. The authors should organization of the text and components in the figures that will make it easier for the reader to connect statements and the data presented in figures/tables. There are also a few discrepancies to be clarified in terms of the experimental approach and analysis as outlined in the specific comments below.

General Comments:

1. Is there a reason to use up to 100 mg/L of the specific substrates? Such as, how environmentally relevant would these concentrations be to understanding impacts within freshwater systems.
2. Figures, methods, and results some times have a different order of the substrates presented. Please check that there is consistency in the text and in the figures/tables being referred to as this presents some confusion when visually comparing the statements and figures.
3. The numbering for the supplementary figures are presented in a different order than mentioned in the text.

Abstract

29: 'significant community growth'? by what statistical analysis was this significant?

31: 'notable enrichment subgroups visible already at 0.1 mg C/l initial compound concentration' Is this for all three compounds? Also, hat about benzoate? There is no mention in the abstract on the results for this compound.

Results:

150-161: Community growth rates increased at up to 10 mg C/L but in 1-octanol, the 100 mg C/L concentration had a larger community size than the 1,000 mg C/L treatment (Fig. 1A). This is not mentioned in the text. Is there an explanation for this? Is it a function of the exposure to octanol or an artifact of the study approach?

187: 'on average 25% more cells' Check this again. Some comparisons are less than 10% different.

190: 'average 10% less cells at the same sampling point' t=2 comparison in the figure shows similar numbers and the t=3 comparison is much less than 10%. Please check this.

205: 'between 22-32% for both compounds' Figures 3 does not show this. Please check these numbers.

225: It is mentioned that the 50% of phenol was not recovered, but the table shows 51.6%.

260: It's mentioned that CellCognize cell types and the 16S rRNA gene amplicon sequence data was used for the community composition analysis. Was this for all substrates? The Methods mentioned 'notable methyl jasmonate and myrcene'. Was the 16S analysis performed on the other substrates but not presented in the manuscript?

284-285: How can this assumption be made using cell type class only and not 16S data? It is unclear if 16S data was used.

298: What about phenol and 1-octanol in terms of this data? Was this not performed? Any specific OTUs?

Methods:

454: Were samples kept on ice during transport? How were water samples stored between field site and lab?

474: Should 'at last' be 'last'?

512: Why not the phenol or 1-octanol treatments?

556: Why not benzoate?

Table 2: Should format as presented in Table 1

Figure 3b: Scale should be changed to show 25% Co₂ as the maximum point just like the other figures.

Figure S1: What does InitialCond and HighConc in the legend mean?

Figure S1, S3 captions should immediately identify that the community data is based on cell type classifications.

Reviewer comments:

We thank both reviewers for their critical and constructive suggestions, which have helped to improve our manuscript. To avoid confusion: please note that line numbers in this rebuttal refer to the revised PDF file with all changes marked.

Reviewer #1 (Comments for the Author):

MSystems 1143-20 describes a systems approach to evaluation of microbial responses to chemical pollutants using a combination of cutting-edge technologies along with more traditional isotopic tracer analyses. The approach is a logical extension of the previous work described in reference 18 and elsewhere. The work seems technically sound and the presentation is clear. A particular strength is the consideration of growth rather than excessive reliance on omics. Minor suggestions for improving the presentation are listed below.

Reply: *We thank the reviewer for the summary and positive statements about our work.*

Specific comments:

1. The criticisms of current standard approaches to biodegradability testing could be dialled back a bit. Or provide examples where the standard approaches have led to bad regulatory decisions that would have been reversed by the approach described here.

Reply: *It was not our intention to criticize ready biodegradability tests too harshly, but to describe their current limitations and potentially, overcome such limitations.*

Action: *We have better specified currently viewed limitations in ready biodegradability tests and added more examples of this in I. 106-110 of the introduction.*

2. The broad applications for microbial ecology could be emphasized more with biodegradability testing for regulatory decisions as one example. It wouldn't hurt to mention the demands of REACH, with the caveat that the strategy described here might not be amenable to high throughput.

Reply and action: *We have added a paragraph in the Introduction (I. 114-125) and Discussion (I. 458-466) to highlight the importance of microbial ecology and of the envisioned FCM method here for biodegradability testing.*

3. Writing is clear, but a few passages should be reworded for simplicity and clarity. Examples include but not limited to: Lines 15-19, 316-319. Minimize "environmental compartment", "benchmarking", "framework", "methodological" and combinations thereof.

Reply and action: *We combed through the manuscript and corrected any passages that seemed unclear. An annotated PDF version of the revised manuscript with all changes highlighted is accompanying the revision.*

4. Line 21. Various

Action: Corrected.

5. Line 48. Faster than what? Why is faster important?

Reply: We thank for this remark as we believe this is a key advantage for flow cytometry methods. Despite the power and accuracy of sequencing-based methods, their processing takes a long time (weeks to months), which precludes routine real-time dynamic monitoring.

Action: We replaced “faster” with “near real-time”. We have further reiterated this aspect in the discussion (l. 461-463 and 597-599).

6. Lines 73-76. Tell the reader why it would be important and how it would change regulatory decisions.

Reply and Action: We have better specified the currently viewed limitations on ready biodegradability tests and explained how FCM-based community tools could be useful for a more integrated view. (l. 119-125, see points 1 and 2). See further l. 459-463 of the revised discussion.

7. Where possible minimize sections in the Results that read the data.

Reply and action: We have combed through the complete results section and remove phrases that too obviously refer to visible results.

8. Lines 311-314, 396-398. Tell the reader what is new.

Reply and Action: We have revised those parts in the discussion (l. 457-466).

9. Lines 331-332. Provide reference to support the statement.

Reply and Action: We explained this statement (>60% mineralization) and provided an appropriate reference (l. 512)

10. Line 408. “..we feel relatively confident...” Replace with confidence intervals or something similar.

Reply and action: Our statement was not meant as a confidence interval, but more as a general conclusion about the benchmarking of the FCM method. We have removed this sentence within the newly added final discussion paragraph (l. 601-615).

11. Lines 499-511. Briefly explain how changes in cell size/shape during the growth cycle were accommodated by the system.

Reply: Briefly, the artificial neural network model was trained with cells from different microbial strains, one of which was under different growth conditions and phases as described in Reference 16. Growth of subgroups (encompassed by the cell type)-in this work was recognized and quantified as an increase of the number of cells falling within the predefined 32 classes.

Action: We rephrased this part (l. 706-710) to make it more clear.

12. Line 439. Likely benzoic acid or sodium benzoate was purchased.

Reply and action: *We used sodium benzoate and not benzoic acid. This was clarified: benzoate → sodium benzoate.*

13. Lines 405-411. Tell the readers about the strengths and weaknesses, barriers to practical application, questions remaining, quantitative comparison with standard methods, potential applications in other areas of microbial ecology. For example, discuss potential problems of cell sorting with sludge/soil, loss of resolution as members of specific populations undergo dramatic changes in cell size and shape during the growth cycle, limitations of amplicon sequencing for following populations of specific degraders.

Reply and action: *We thank the reviewer for this suggestion and have included a paragraph on this in the discussion (l. 601-615). To explain all the suggested aspects in great detail would be out of scope of the manuscript and more suitable for a mini-review.*

Reviewer #2 (Comments for the Author):

The work presented is a novel approach to understanding biodegradation of organics by identifying cell types using a technique developed by the authors (the CellCognize pipeline) to provide a higher resolution in identifying microbial community changes. Data and the approach summarized could be highly impactful in identifying populations in mixed communities that are impacted by xenobiotics and the interrelationship to transformation of organic substrates. The authors used phenol, 1-octanol, and benzoate as the reference compounds and compared this to three others with two compounds with intermediate biodegradability (in addition to a non-biodegradable control). The authors determined growth rates of a freshwater microbial community when using the reference substrates as well as the incorporation of C14-labelled carbon into the biomass and evolution to carbon dioxide. Subgroups that were enriched by substrates were identified based on the CellCognize pipeline and the 16S amplicon data. Using flow cytometry, this approach will substantially change how we understand responses of complex bacterial communities to substrates. The authors should organization of the text and components in the figures that will make it easier for the reader to connect statements and the data presented in figures/tables. There are also a few discrepancies to be clarified in terms of the experimental approach and analysis as outlined in the specific comments below.

Reply: *We thank the reviewer for the summary and positive statements about our work. We have revisited the figure and table components, as recommended.*

General Comments:

1. Is there a reason to use up to 100 mg/L of the specific substrates? Such as, how environmentally relevant would these concentrations be to understanding impacts within freshwater systems.

Reply: As we wanted to benchmark the method, we decided to use a relatively broad range of concentrations for the three readily biodegradable compounds, in order to show the detection limit as well as a toxicity range. Also, we considered the recommended concentration range for OECD ready biodegradability tests which is 2–100 mg C l⁻¹ as mentioned in I. 70.

Action: We explained this in I. 108, I.178-180 and I 493-499.

2. Figures, methods, and results some times have a different order of the substrates presented. Please check that there is consistency in the text and in the figures/tables being referred to as this presents some confusion when visually comparing the statements and figures.

Reply and action: We have verified that in all figures and tables the elements 'benzoate, phenol and 1-octanol' are in the same order.

3. The numbering for the supplementary figures are presented in a different order than mentioned in the text.

Reply and action: We have verified once more that all supplementary figures are properly listed and mentioned.

Abstract

29: 'significant community growth'? by what statistical analysis was this significant?

Reply and action: The statistics was described in the main text and table 1 (I. 977). We specified the growth range here in the abstract (I. 23-24).

31: 'notable enrichment subgroups visible already at 0.1 mg C/l initial compound concentration' Is this for all three compounds? Also, hat about benzoate? There is no mention in the abstract on the results for this compound.

Reply and action: We specified that this was for phenol and 1-octanol.

Results:

150-161: Community growth rates increased at up to 10 mg C/L but in 1-octanol, the 100 mg C/L concentration had a larger community size than the 1,000 mg C/L treatment (Fig. 1A). This is not mentioned in the text. Is there an explanation for this? Is it a function of the exposure to octanol or an artifact of the study approach?

Reply: We stated that this is an effect of toxicity (like for phenol at the highest concentration). This was specified in I. 208 "...indicating toxicity and growth inhibition".

187: 'on average 25% more cells' Check this again. Some comparisons are less than 10% different.

Reply and action: The observed differences between pasteurized and non-pasteurized at the same time point were: 29%, 28%, and 21% for t1, t2 and t6. We specified this range in the text in I. 281.

190: 'average 10% less cells at the same sampling point' t=2 comparison in the

figure shows similar numbers and the t=3 comparison is much less than 10%. Please check this.

Reply and action: *The observed differences between pasteurized and non-pasteurized with phenol at the same time point were: 80%, 0% and 33%, for t1, t2 and t6, respectively. We specified the range in the revised text in I. 284.*

205: 'between 22-32% for both compounds' Figures 3 does not show this. Please check these numbers.

Reply: *This is a combination of the data from the Table 1 and Figure 3 (as we specify in I. 205, 206). 32% comes from the 1-octanol data in Table 1, 22% is from the phenol experiment of Figure 3a.*

225: It is mentioned that the 50% of phenol was not recovered, but the table shows 51.6%.

Reply: *We thank the reviewer for pointing this out. These statements referred to Table 1 where an upper and lower range of the data are presented.*

Action: *We have changed this for the actual range of measured values instead of a broadly rounded value (I. 321-322).*

260: It's mentioned that CellCognize cell types and the 16S rRNA gene amplicon sequence data was used for the community composition analysis. Was this for all substrates? The Methods mentioned 'notable methyl jasmonate and myrcene'. Was the 16S analysis performed on the other substrates but not presented in the manuscript?

Reply: *16S rRNA gene amplicon analysis was conducted for phenol, 1-octanol, myrcene and methyl jasmonate (and no-carbon control) – as shown in Figure 6b, but sequencing data from phenol and 1-octanol incubations had been presented in our previous work (reference 16).*

Action: *We specified in I. 731-733 that phenol and 1-octanol data have been presented in our previous work; Bioproject Nr PRJNA641590.*

284-285: How can this assumption be made using cell type class only and not 16S data? It is unclear if 16S data was used.

Reply: *We thank the reviewer for this remark. We use the FCM data here because of the higher time resolution – actually as a point to show how such denser time kinetic data can be important. The extreme enrichment seen in phenol and 1-octanol at the final time point has been described in our previous work (reference 16).*

Action: *We added the reference and statement (I. 444-448).*

298: What about phenol and 1-octanol in terms of this data? Was this not performed? Any specific OTUs?

Reply and action: *We included the corresponding data for phenol and 1-octanol in Figure 6c, and the OTU lists in Table S3. We further added corresponding text (I. 444-448).*

Methods:

454: Were samples kept on ice during transport? How were water samples stored between field site and lab?

Reply and action: *Transportation was completed within 20 min. The water samples were not stored at all. The bacterial suspension was prepared and used immediately after transport and filtering. We specified this in the methods section in I 661.*

474: Should 'at last' be 'last'?

Reply and action: *'at' deleted.*

512: Why not the phenol or 1-octanol treatments?

Reply and action: *As mentioned above, incubations with phenol and 1-octanol were also used for 16S rRNA community sequencing data. Please see the revised Figure 6c and Table S3.*

556: Why not benzoate?

Reply and action: *Benzoate was not included here because we had two other readily biodegradable substances (phenol and 1-octanol). Hence, we refrained from repeating the 14C experiments with benzoate.*

Table 2: Should format as presented in Table 1.

Reply and action: *We revised the header row of Table 2.*

Figure 3b: Scale should be changed to show 25% Co₂ as the maximum point just like the other figures.

Reply and action: *Scale was adjusted.*

Figure S1: What does InitialCond and HighConc in the legend mean?

Reply and action: *This refers to the different set of experiments as specified in Figure S1 caption as well as in Table S1.*

Figure S1, S3 captions should immediately identify that the community data is based on cell type classifications.

Reply and action: *Figure captions were revised correspondingly by specifying "Community composition changes from FCM-CellCognize cell-type deconvolutions"*

.

January 15, 2021

Prof. Jan Roelof van der Meer
University of Lausanne
Department of Fundamental Microbiology
Batiment Biophore
Quartier Unil-Sorge
Lausanne CH 1015
Switzerland

Re: mSystems01143-20R1 (Assessing biodegradability of chemical compounds from microbial community growth using flow cytometry)

Dear Prof. Jan Roelof van der Meer:

Thank you for returning your revised manuscript for consideration. You have adequately addressed all reviewer concerns, and I am delighted to accept your manuscript for publication in MSystems.

Your manuscript has been accepted, and I am forwarding it to the ASM Journals Department for publication. For your reference, ASM Journals' address is given below. Before it can be scheduled for publication, your manuscript will be checked by the mSystems senior production editor, Ellie Ghatineh, to make sure that all elements meet the technical requirements for publication. She will contact you if anything needs to be revised before copyediting and production can begin. Otherwise, you will be notified when your proofs are ready to be viewed.

Sincerely,

Laura Hug
Editor, mSystems

Journals Department
Table S1: Accept
Table S3: Accept
Fig. S5: Accept
Fig. S4: Accept
Fig. S3: Accept
Table S2: Accept
Fig. S1: Accept
Fig. S2: Accept
Table S4: Accept